# Long-Term Neuroradiological and Clinical Evaluation of NBIA Patients Treated with a Deferiprone Based Iron-Chelation Therapy

**DOI:** 10.3390/jcm11154524

**Published:** 2022-08-03

**Authors:** Nicola Romano, Giammarco Baiardi, Valeria Maria Pinto, Sabrina Quintino, Barbara Gianesin, Riccardo Sasso, Andrea Diociasi, Francesca Mattioli, Roberta Marchese, Giovanni Abbruzzese, Antonio Castaldi, Gian Luca Forni

**Affiliations:** 1Department of Diagnostic and Interventional Neuroradiology, EO Ospedali Galliera, 16128 Genoa, Italy; nicola.romano@galliera.it (N.R.); riccardo.sasso@galliera.it (R.S.); antonio.castaldi@galliera.it (A.C.); 2Clinical Pharmacology Unit, EO Ospedali Galliera, 16128 Genoa, Italy; giammarco.baiardi@galliera.it (G.B.); fmattiol@unige.it (F.M.); 3Department of Internal Medicine, Pharmacology & Toxicology Unit, University of Genoa, 16132 Genoa, Italy; 4Centro della Microcitemia, delle Anemie Congenite e dei Disordini del Metabolismo del Ferro, EO Ospedali Galliera, 16128 Genoa, Italy; valeria.maria.pinto@galliera.it (V.M.P.); sabrina.quintino@galliera.it (S.Q.); barbara.gianesin@galliera.it (B.G.); 5Department of Health Sciences (DISSAL), Radiology Section, University of Genoa, 16132 Genoa, Italy; andrea.diociasi@gmail.com; 6IRCCS Policlinico San Martino, 16132 Genoa, Italy; rmarchese@neurologia.unige.it; 7Clinical Neurophysiology, Department of Neurosciences, Ophthalmology and Genetics, University of Genoa, 16132 Genoa, Italy; giabbr@unige.it

**Keywords:** NBIA, deferiprone, brain iron, iron overload, neurodegeneration

## Abstract

Neurodegeneration with brain iron accumulation (NBIA) comprises various rare clinical entities with brain iron overload as a common feature. Magnetic resonance imaging (MRI) allows diagnosis of this condition, and genetic molecular testing can confirm the diagnosis to better understand the intracellular damage mechanism involved. NBIA groups disorders include: pantothenate kinase-associated neurodegeneration (PKAN), mutations in the gene encoding pantothenate kinase 2 (PANK2); neuroferritinopathy, mutations in the calcium-independent phospholipase A2 gene (PLA2G6); aceruloplasminemia; and other subtypes with no specific clinical or MRI specific patterns identified. There is no causal therapy, and only symptom treatments are available for this condition. Promising strategies include the use of deferiprone (DFP), an orally administered bidentate iron chelator with the ability to pass through the blood–brain barrier. This is a prospective study analysis with a mean follow-up time of 5.5 ± 2.3 years (min–max: 2.4–9.6 years) to define DFP (15 mg/kg bid)’s efficacy and safety in the continuous treatment of 10 NBIA patients through clinical and neuroradiological evaluation. Our results show the progressive decrease in the cerebral accumulation of iron evaluated by MRI and a substantial stability of the overall clinical neurological picture without a significant correlation between clinical and radiological findings. Complete ferrochelation throughout the day appears to be of fundamental importance considering that oxidative damage is generated, above, all by non-transferrin-bound iron (NTBI); thus, we hypothesize that a (TID) administration regimen of DFP might better apply its chelating properties over 24 h with the aim to also obtain clinical improvement beyond the neuroradiological improvement.

## 1. Introduction

Iron metabolism is finely regulated in humans because there is no physiological pathway for the efficient excretion of excesses; thus, iron overload (IO) leads to progressive deposition in tissues and organ damage [1]. Through the Fenton reaction, excess iron generates reactive oxygen species (ROS), which can increase oxidative stress and co-participate in neurodegeneration in some genetic disorders [2,3]. Neurodegeneration with brain iron accumulation (NBIA) comprises various clinical entities with brain IO as a common feature [4]. Magnetic resonance imaging (MRI) allows premortem diagnosis of this condition demonstrating brain iron accumulation using sequences sensitive to metallic content, as gradient-echo T2*-weighted images [5,6,7,8,9]; genetic molecular testing can now pose confirmation to better clear the intracellular damage mechanism involved [10].

An MRI may help to distinguish subtypes of NBIA characterized by iron deposition. An algorithm proposed by Kruer et al. [11] suggested considering four features when evaluating patients with suspected NBIA, in particular, the presence of: (a) the “eye of the tiger sign” (with bilateral low signal T2-weighted images in the globus pallidus (due to iron accumulation) with a central high signal (due to gliosis and spongiosis) typical of pantothenate kinase-associated neurodegeneration (PKAN); (b) T2 hypointensity of the caudate, putamen, and/or thalamus, which may suggest a diagnosis of neuroferritinopathy, Kufor–Rakeb syndrome (KRS) and aceruloplasminemia (ACP); (c) T2 white matter hyperintensity that may be associated with Woodhouse–Sakati syndrome (WSS), fatty acid hydroxylase-associated neurodegeneration (FAHN), and neuroaxonal dystrophy (NAD); and (d) T1 substantia nigra hyperintensity typical of static encephalopathy in childhood with neurodegeneration in adulthood (SENDA).

NBIA is a heterogeneous family of disorders including: PKAN caused by mutations in the gene encoding pantothenate kinase 2 (PANK2); neuroferritinopathy caused by a mutation in the ferritin light chain gene (FTL); NBIA type 2, associated with mutations in the calcium-independent phospholipase A2 gene (PLA2G6); ACP, associated with mutations in the ceruloplasmin gene (CP); and other subtypes with no specific clinical or MRI specific patterns identified [12,13].

Until now, there is no causal therapy and only symptom treatments are available for this condition. Promising strategies include the use of deferiprone (DFP), an orally administered bidentate iron chelator with the ability to pass through the blood–brain barrier (BBB) [14,15], deemed safe and effective in reducing brain IO in NBIA patients [16]. DFP (15 mg/kg bid) is able to act as a reverse siderophore through the BBB and improve neurological manifestations when administered at an early symptomatic stage [17] or, nonetheless, slow down disease progression [18,19].

This is a prospective study analysis with a mean follow-up time of 5.5 ± 2.3 years (min–max: 2.4–9.6 years) to define DFP (15 mg/kg bid) efficacy and safety in the continuous treatment of NBIA patients through clinical and neuroradiological MRI evaluation.

## 2. Materials and Methods

### 2.1. Patients and Methods

After the conclusion of the first one-year pilot study [17] and the first follow-up period [16] some of the patients who have been treated in our Centre since 2008 on average are still in therapy and monitored. Of the patients enrolled in the previous clinical trials [16,17,19,20], 10 patients were still in treatment and were included in this analysis (4 with PKAN, 2 PLAG2G6, 1 with neuroferritinopathy, and 3 with idiopathic NBIA). All 10 patients included in this study received DFP solution (Apopharma, Toronto, ON, Canada) at 15 mg/kg BID. All participants gave written informed consent. For procedures and safety monitoring see ClinicalTrials.gov (Identifier: NCT00907283) [20].

Deferiprone (DFP) is an iron chelator, which is approved [21] for the treatment of iron overload due to transfusion in the thalassemia at a total daily dose of 75 mg/kg divided into three administrations (TID). It was chosen because, at present, this is the only approved iron chelator able to cross the blood–brain barrier. The 15 mg/kg BID regimen used to treat our patients was initially adopted by Boddaert et al. [22] for the treatment of Freidreich’s Ataxia, a condition in which decreased iron sulfur clustering and heme formation lead to mitochondrial iron accumulation, with ensuing oxidative damage that primarily affects the sensory neurons, but also the myocardium, and the endocrine glands. Boddaert’s team first tried a dose of 80 mg/kg, which was used in thalassemia for iron overload. Unfortunately, the appearance of several side effects in one patient led to the 15 mg/kg BID regimen adoption with the aim of reducing toxicity. This regimen of 15 mg/kg BID, with a total daily dose of 30 mg/kg, is then deemed suitable for reducing iron accumulation in specific brain areas without interfering with hematological parameters in patients with no systemic iron overload (due to frequent blood transfusion as in Thalassemia Major).

Follow-up hematological visits were performed every 7 days for the first year and then every four months to assess the safety of the treatment and clinical outcomes. MRI evaluations were performed yearly to examine the IO on the globus pallidus internus (GPi) nuclei through the T2* parameter; at the same time, a neurological clinical evaluation was conducted. The relaxation rate R2* (1/T2*) was used to investigate the variation in iron overload during the follow-up period.

Patients underwent a brain MRI on a 1.5-T (Achieva, Philips Healthcare, Best, the Netherlands) scanner using the same acquisition protocol, including (i) a 3D-FLAIR sequence (repetition time (TR) 4800 ms, echo time (TE) 306.259 ms, inversion time (IT) 1660 ms; flip angle: variable; slice thickness 1.12 mm); (ii) Spin-Echo T2-weighted sequence (TR 5043.37 ms, TE 100 ms, flip angle 90°, slice thickness 5 mm); (iii) Spin-Echo T1-weighted sequence (TR 4994 ms, TE 15 ms, flip angle: 25°, slice thickness 1.12 mm); (iv) multi_Echo T2 images (TR 2892.74 ms, TE 20 ms-40 ms-60 ms-80 ms-100 ms, Flip Angle 90°); smFFE sequence (TR 778.4 ms, TE 4.604 ms, slice thickness 5mm); and (v) ven-Bold (TR 34.9782 ms, TE 49.979 ms, flip angle 15°, slice thickness 2 mm).

Two neuroradiologists reviewed the MRI images to provide a qualitative evaluation based on the appropriate analysis of a priori defined regions of interest (ROI). Quantitative assessment of brain iron was performed with T2* relaxometry, using the axial multi_Echo T2 sequence and quantitative T2* maps were calculated off-line using a custom-made reconstruction algorithm (FuncTool v. 5.2.09, GE Medical Systems). ROI were manually drawn by a single neuroradiologist (on T2* maps).

For the clinical tests, the patients were evaluated by neurologists expert in movement disorders using the Unified Parkinson’s Disease Rating (UPDRS/III–Motor Section), ICARS, and the Unified Dystonia Rating (UDRS) scales, that were administered at baseline and during follow-up.

### 2.2. Statistical Analysis

Descriptive statistics were performed with means and standard deviations (SD), counts, and proportions. R2* measurements as function of time were displayed by scatterplots, and boxplots were used to present R2* reduction in different sites between basal and last follow-up. The association between R2* and time was studied for each patient with Pearson’s correlation coefficient, after the visualization of two-way scatterplots of the data. Pairwise comparisons (Bonferroni-corrected), of R2* were used to assess differences between groups (basal–follow up, left GPi–right GPi). All of the tests were two-sided with a type 1 error (α) of 0.05; the analyses were conducted using R Core Team (2021) (R Foundation for Statistical Computing, Vienna, Austria. URL https://www.R-project.org/).

## 3. Results

The clinical and diagnostic characteristics of the 10 patients included in the analysis are reported in Table 1. Three patients were males and seven were females; the mean age at basal was 41 ± 21 years. The mean number of MRI evaluations for each patient was 6.0 ± 2.4, and the mean follow-up time was 5.5 ± 2.3 years (min–max: 2.4–9.6 years).

No hematological severe toxicity was registered; although 7/10 patients needed iron oral supplementation in order to maintain ferritin values > 30 µg/L. Independent neurological clinical evaluation at each follow-up visit proved the clinical stability of the disease in 50% of cases.

The temporal variations of R2* in the left and right GPi are reported for each patient in Figure 1A,B. A high negative correlation between R2* and time (Table 1) was observed in eight patients. In two patients, the iron overload showed a stability during the entire follow-up period with a negative correlation coefficient Rp that did not reach the level of significance.

In the left GPi the R2* decreased from 47.6 ± 6.4 Hz to 37.3 ± 5.8 Hz (*p* < 0.0001) and in the right GPi, it decreased from 48.4 ± 6.2 Hz to 37.9 ± 6.6 Hz (*p* < 0.0001); no significant differences were observed between left and right GPi (Figure 2).

## 4. Discussion

NBIA syndromes encompass a wide range of clinical and molecular features [13]. The more frequent alteration correlated with NBIA is the deficiency of PANK2 and, consequently, of CoA, which leads to the extracellular accumulation of a series of intermediate metabolites, such as cysteine and pantethine. A high concentration of cysteine is found in the patients’ globus pallidus where it undergoes rapid self-oxidation with abundant production of free radicals in the presence of iron [23].

There is a strong connection between the alteration in CoA levels, ROS production and apoptosis; the PKAN’s pathomechanism is directly related to the overproduction of ROS and unbalanced mitochondrial redox, which may trigger a neuronal death cascade [24]. Thus, it is thought iron-induced lipid peroxidation could be responsible for the cellular damage observed in affected NBIA patients.

Our results show a substantial stability of the overall clinical neurological picture, despite the progressive decrease in the cerebral accumulation of iron. Therefore, during follow-up periods, we did not observe a significant correlation between clinical and radiological findings. Though noteworthy, this might be due to the limited sample size of the study population, because NBIA has an estimated prevalence of 1–9/1,000,000 in the general population, and thus may be considered an ultra-orphan disease.

Our data confirm the results described in the pilot studies [18,19] and are consistent with those of Klopstock et al. [25], the only randomized clinical trial comparing deferiprone to placebo, where a predefined subgroup analysis separately assessing patients with classic or atypical PKAN provided some evidence to show that deferiprone use in patients with atypical PKAN could be associated with slower progression of clinical symptoms.

We hypothesize this might be caused by the peroxidative damage linked to the progressive accumulation of iron, which is irreversible if not diagnosed and treated early [17]. In fact, the iron chelating treatment with deferiprone 15 mg/kg BID can prevent further clinical deterioration, as supported by the stabilization of the GPi iron overload we observed in this cohort of treated patients. Otherwise, Roa-Sanchez et al.’s [26] study showed that, without a pharmacological treatment, the clinical course of this pathological cascade is the progressive atrophy of the GPi nuclei and clinical neurological deterioration; a volume loss in the globus pallidus of 1.35 ± 2.79% per year of the FU was statistically significantly related to the progression of dystonia

Therefore, guaranteeing complete ferrochelation throughout the day appears to be of fundamental importance considering that oxidative damage is generated above all by nontransferrin-bound iron (NTBI); a highly reactive Fe^+2^ subspecies of NTBI called labile plasma iron (LPI) can enter cells through ion transporters that are normally designed to carry other divalent cations. These ion transporters are generally not regulated by intracellular iron concentration. Thus, bypassing the TfR1 regulatory mechanisms, iron loading proceeds even when cytosolic iron levels are very high. However, NTBI/LPI levels can fall to zero as soon as there is a circulating chelator, blocking further loading through ion transporters. This means a chelator should be circulating at all times [27].

Deferiprone has a short half-life of 2–3 h. So, we hypothesize that a further split-dosage, more than BID, could better show the chelating properties of deferiprone in our study population without worsening the safety profile of the drug, as the total daily dose would remain the same (30 mg/kg/d). This would also be in accordance with the three times a day (TID) administration regimen for the chelation treatment of iron overload in transfusion dependent thalassemia [21].

## 5. Conclusions

NBIA-related GPi iron overload is safely reversible by a deferiprone 15 mg/kg BID iron-chelating treatment; however, clinical neurological manifestations of the disease are slowed down. We hypothesized that a further split administration regimen of DFP might better apply its chelating properties over 24 h with the aim of also obtaining clinical improvement.

## Figures and Tables

**Figure 1 jcm-11-04524-f001:**
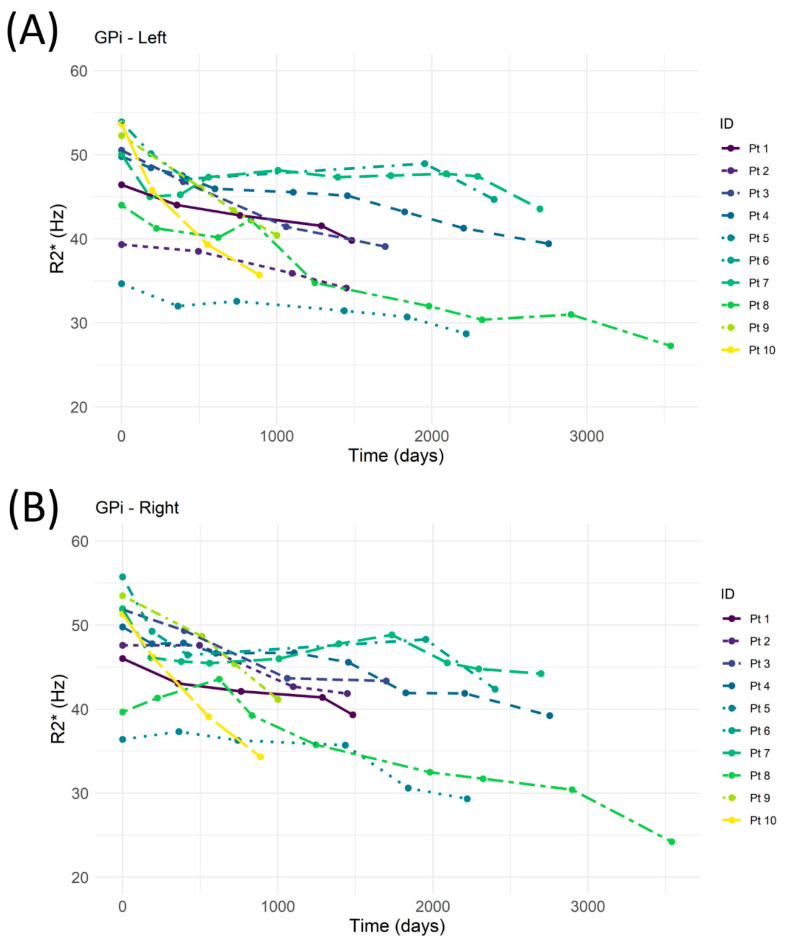
MRI-R2* evaluation of left (**A**) and right (**B**) globus pallidus internus (GPi) as a function of time. Pt (patient).

**Figure 2 jcm-11-04524-f002:**
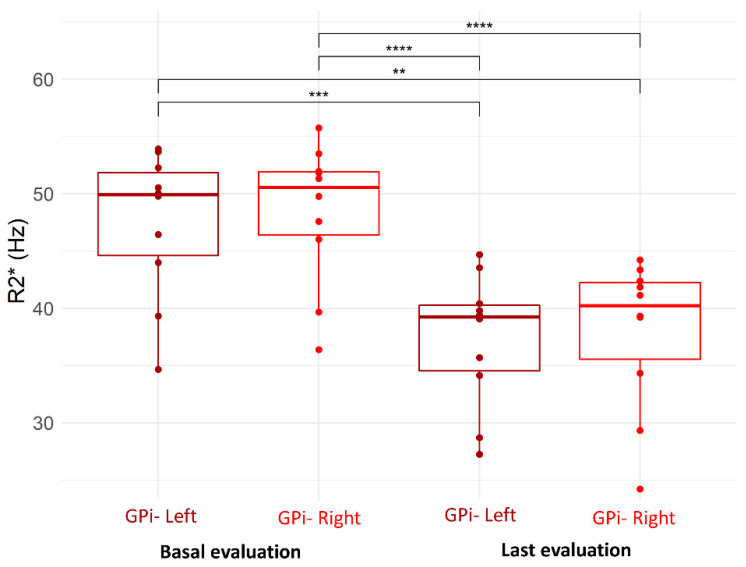
MRI-R2* of left and right globus pallidus internus (GPi)—comparison between basal and last evaluation (“**” *p* ≤ 0.01; “***” *p* ≤ 0.001; “****” *p* ≤ 0.0001).

**Table 1 jcm-11-04524-t001:** Clinical and diagnostic characteristics.

Gender	Age yrs	Diagnosis	N. of MRI	Follow-Up (years)	MRI R2* (Hz) GPi Left			MRI R2* (Hz) GPi Right			MRI-Details	Clinical Evaluation at LAST MRI FU
					Basal	Last	Rp ^#^	Basal	Last	Rp ^#^		
F	53.5	Neuro ferritinopathy	5	4.1	46.4	39.8	−0.98 (**)	46.0	39.3	−0.95 (*)	T2* hypointensities GPi(bilateral), NR(bilateral)	Stable with no progression
M	65.9	NBIA	4	4.0	39.3	34.1	−0.98 (*)	47.6	41.9	−0.94 (·)	T2* hypointensities GPi(bilateral), ND(bilateral)	Stable with no progression
F	72.1	NBIA	4	4.7	50.5	39.1	−0.98 (*)	51.9	43.4	−0.95 (·)	T2* hypointensities GPi(bilateral), SN	Stable with no progression
M	41.8	PKAN	9	7.5	49.8	39.4	−0.98 (****)	49.8	39.2	−0.97 (****)	T2* hypointensities in GPi(bilateral)—‘tiger eye’ sign	Definite worsening with severe dysarthria, instability and falls
M	54.0	NBIA	6	6.1	34.7	28.7	−0.92 (**)	36.4	29.3	−0.88 (*)	T2* hypointensities in GPi (bilateral)	Improvement
F	17.8	PKAN	5	6.6	53.9	44.7	−0.71	55.8	42.4	−0.72	T2* hypointensities in GPi (bilateral), SN	Rapid and severe global worsening
F	25.8	PLAG2G6	10	7.4	50.0	43.6	−0.26	52.0	44.2	−0.48	T2* hypointensities in GPi (bilateral)	Definite worsening with severe dystonia, instability and falls
F	40.8	PKAN	9	9.7	44.0	27.3	−0.95 (****)	39.7	24.2	−0.95 (****)	T2* hypointensities in GPi (bilateral)—‘tiger eye’ sign	Stable with limited progression
F	20.3	PKAN	4	2.7	52.3	40.4	−0.99 (***)	53.5	41.2	−0.99 (**)	T2* hypointensities in GPi (bilateral)	Definite worsening with gait difficulties and disarthria
F	14.7	PLAG2G6	4	2.4	53.7	35.7	−0.97 (*)	51.3	34.3	−0.99 (**)	T2* hypointensities in GPi (bilateral)	Gradual worsening

^#^ Rp: Pearson’s correlation coefficient; “·” *p* ≤ 0.1; “*” *p* ≤ 0.05; “**” *p* ≤ 0.01; “***” *p* ≤ 0.001; “****” *p* ≤ 0.0001; F (female); M (male); yrs (years); MRI (magnetic resonance imaging), GPi (globus pallidus internus), NR (nucleus ruber), ND (nucleus dentatus), SN (substantia nigra).

## Data Availability

Data are available on request to the authors.

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
