# Peer review of "Long-Term Neuroradiological and Clinical Evaluation of NBIA Patients Treated with a Deferiprone Based Iron-Chelation Therapy"

_jcm, 2022, doi:10.3390/jcm11154524_

Round 1

Reviewer 1 Report

Page 2, lines 47 and 48

Magnetic resonance imaging (MRI) allows premortem diagnosis of this condition (detail following the informations below)

All of the NBIA disorders feature iron deposition in the globus pallidus but differ in the co-occurrence of other findings. They are unified by the clinical constellation of a movement disorder and neurodegenerative course. All are autosomal recessive except for neuroferritinopathy.

MR imaging is of tremendous utility in the evaluation of brain iron disorders and facilitates clinical diagnosis. Despite its usefulness as a biomarker, the pathophysiologic role of iron deposition in NBIA remains uncertain. Associated MR imaging abnormalities may help to distinguish subtypes of NBIA and facilitate a more definitive diagnosis

See Table 2 of the article Neuroimaging Features of Neurodegeneration with Brain Iron Accumulation

M.C. Kruer, N. Boddaert, S.A. Schneider, H. Houlden, K.P. Bhatia, A. Gregory, J.C. Anderson, W.D. Rooney, P. Hogarth, S.J. Hayflick

American Journal of Neuroradiology Sep 2011, DOI: 10.3174/ajnr.A2677

include the paper in the presentation of MRI findings

Punaro E, Feltrin KY, Reis F. Magnetic resonance imaging in acaeruloplasminaemia - a rare differential diagnosis of microcytic anaemia with iron overload. Br J Haematol. 2017 Sep;178(6):837. doi: 10.1111/bjh.14816.

Line 136 page 6

NBIA syndromes encompass a wide range of clinical and molecular features (11)

patients' Globus pallidus line 140 page 6, substitute by, patients’ globus pallidus

in discussion, include the limitation of the small number of subjects

compare your results with other studies in the literature, including the follow-up results in different papers.

Author Response

Response to editors & reviewers

“Long-term neuroradiological and clinical evaluation of NBIA patients treated with a deferiprone based iron chelation therapy (jcm-1819708)

Dear Editor,

We really appreciated the detailed work the Reviewers did on our paper. We have consequently reviewed the manuscript according to the useful suggestions of the Reviewers (see manuscript with tracked changes).

REVIEWER n1.

Q1:

Page 2, lines 47 and 48

Magnetic resonance imaging (MRI) allows premortem diagnosis of this condition (detail following the informations below)

All of the NBIA disorders feature iron deposition in the globus pallidus but differ in the co-occurrence of other findings. They are unified by the clinical constellation of a movement disorder and neurodegenerative course. All are autosomal recessive except for neuroferritinopathy.

MR imaging is of tremendous utility in the evaluation of brain iron disorders and facilitates clinical diagnosis. Despite its usefulness as a biomarker, the pathophysiologic role of iron deposition in NBIA remains uncertain. Associated MR imaging abnormalities may help to distinguish subtypes of NBIA and facilitate a more definitive diagnosis

See Table 2 of the article Neuroimaging Features of Neurodegeneration with Brain Iron Accumulation

M.C. Kruer, N. Boddaert, S.A. Schneider, H. Houlden, K.P. Bhatia, A. Gregory, J.C. Anderson, W.D. Rooney, P. Hogarth, S.J. Hayflick

American Journal of Neuroradiology Sep 2011, DOI: 10.3174/ajnr.A2677

include the paper in the presentation of MRI findings

Punaro E, Feltrin KY, Reis F. Magnetic resonance imaging in acaeruloplasminaemia - a rare differential diagnosis of microcytic anaemia with iron overload. Br J Haematol. 2017 Sep;178(6):837. doi: 10.1111/bjh.14816.

A1: We modified the manuscript and added the suggested references 

Q2:

Line 136 page 6 NBIA syndromes encompass a wide range of clinical and molecular features (11)

patients' Globus pallidus line 140 page 6, substitute by, patients’ globus pallidus

A2: we modified the manuscript as suggested.

Q3:

in discussion, include the limitation of the small number of subjects.

A3: as suggested, we included limitation of the study to the paper in lines from 158 to 160.

Q4:

compare your results with other studies in the literature, including the follow-up results in different papers.

A4: done as suggested in lines from 168 to 173.

We sincerely thank the Reviewers for appreciating our work.

Reviewer 2 Report

Manuscript deals with NBIA (neurodegeneration with brain iron accumulation) patients which were examined before and after deferiprone (bidendate iron chelator) administration using clinical tests and MRI. Results of the study showed that iron overload could be reversible and clinical manifestations could be slowed down by using deferiprone 15 mg/kg two times a day. The study is of great scientific and clinical interest but however there are some important issues that should be concerned. 

The most important issue that should be thoroughly revised is MRI protocol description. It is recommended to provide more complete sequences description including slice thickness, flip angle, dist factor, examination time etc., e.g. IT for used T1-weighted sequence which seems to be IR sequence, different echo-times for multi-echo T2 (T2*?) or delta echo time, moreover IT in 3D-FLAIR could contain an error or please explain how it is possible that IT>TR.

According to materials and methods section deferiprone was administered two times a day, but in conclusion authors speculate about using it three times a day (why not more often?) which is obviously logical regarding constant concentration maintenance but is not supported by any received data.

To end with, manuscript should be revised regarding English language usage, paper contains also simple mistakes, the authors are kindly asked to pay special attention to lines 47, 51, 75, 76, 80, 104, 109, 143 (pathogenesis would sound better) and others.

Author Response

Response to editors & reviewers

“Long-term neuroradiological and clinical evaluation of NBIA patients treated with a deferiprone based iron chelation therapy (jcm-1819708)

 Dear Editor,

We really appreciated the detailed work the Reviewers did on our paper. We have consequently reviewed the manuscript according to the useful suggestions of the Reviewers (see manuscript with tracked changes).

REVIEWER n2.

Q1:

The most important issue that should be thoroughly revised is MRI protocol description. It is recommended to provide more complete sequences description including slice thickness, flip angle, dist factor, examination time etc., e.g. IT for used T1-weighted sequence which seems to be IR sequence, different echo-times for multi-echo T2 (T2*?) or delta echo time, moreover IT in 3D-FLAIR could contain an error or please explain how it is possible that IT>TR.

A1: We modified the manuscript improving the MRI protocol description.

Q2:

According to materials and methods section deferiprone was administered two times a day, but in conclusion authors speculate about using it three times a day (why not more often?) which is obviously logical regarding constant concentration maintenance but is not supported by any received data.

A2:  Deferiprone (DFP) is an iron chelator which is approved (https://www.ema.europa.eu/en/documents/product-information/ferriprox-epar-product-information_it.pdf)  for the treatment of iron overload due to transfusion in Thalassemia at a total daily dose of 75 mg/kg divided into three administrations (TID). It was chosen because, at present, this is the only approved iron chelator able to cross the blood–brain barrier. The 15 mg/kg BID regimen, used to treat our patients, was initially adopted by Boddaert et al. (Boddaert N, Le Quan Sang KH, Rotig A, et al. Selective iron chelation in Friedreich ataxia: biologic and clinical implications. Blood 2007;110:401–408.) for the treatment of Freidreich’s Ataxia, a condition in which decreased iron–sulphur cluster and heme formation lead to mitochondrial iron accumulation, with ensuing oxidative damage that primarily affects sensory neurons, but also the myocardium, and the endocrine glands. Boddaert’s team first tried a dose of 80 mg/kg, which was used in thalassemia for iron overload. Unfortunately, the appearance of several side effects in one patient, led to the 15 mg/kg BID regimen adoption with the aim of reducing toxicity. This regimen of 15 mg/kg BID, with a total daily dose of 30 mg/kg, is then deemed suitable for reducing iron accumulation in specific brain areas without interfering with haematological parameters in patients with no systemic iron overload (due to frequent blood transfusion as in Thalassemia Major). We then hypothesize that a further splitted-dosage could better show the chelating properties of deferiprone in our study population without worsening the safety profile of the drug, as the total daily dose would remain the same (30 mg/kg/d). This TID regimen would then be similar to the TID regimen of deferiprone in Thalassemia Major with better adherence to the pharmacokinetic profile of the drug.

Q3:

To end with, manuscript should be revised regarding English language usage, paper contains also simple mistakes, the authors are kindly asked to pay special attention to lines 47, 51, 75, 76, 80, 104, 109, 143 (pathogenesis would sound better) and others.

We also have made minor changes to the paper:

Line 5: affiliations of the 2nd Author as those of the 9th Author

Line 47 and 51: misplacing error

A3: Updating of the bibliography on the basis of new added references. In agreement with the Editor, there will be English revised language after the manuscript will be accepted.

We sincerely thank the Reviewers for appreciating our work.

Round 2

Reviewer 2 Report

Manuscript deals with NBIA (neurodegeneration with brain iron accumulation) patients which were examined before and after deferiprone (bidendate iron chelator) administration using clinical tests and MRI. Results of the study showed that iron overload could be reversible and clinical manifestations could be slowed down by using deferiprone 15 mg/kg two times a day.

Manuscript obviously benefited from editing performed by the outhors, however I would like to point out some minor issues.

Line 131: please provide IT, otherwise the sequence can't have T1 contrast

Line 146: Which statistical software was used? Why Pearson's correlation coefficient and not Spearman rank correlation coefficient was used ? 

Line 150: time was studied

Line 153: two-sided

Author Response

We thank the reviewer for the suggestions that we have included in the manuscript

Manuscript deals with NBIA (neurodegeneration with brain iron accumulation) patients which were examined before and after deferiprone (bidendate iron chelator) administration using clinical tests and MRI. Results of the study showed that iron overload could be reversible and clinical manifestations could be slowed down by using deferiprone 15 mg/kg two times a day.

Manuscript obviously benefited from editing performed by the outhors, however I would like to point out some minor issues.

Q1 Line 131: please provide IT, otherwise the sequence can't have T1 contrast

R1: We used Spin-Ehco T1-weighted images so only TR and TE are specified. We added “Spin-Echo” in the text.

Q2 Line 146: Which statistical software was used? Why Pearson's correlation coefficient and not Spearman rank correlation coefficient was used? 

R2: The statistical software used in the analysis was R Core Team (2021). R: A language and environment for   statistical computing. R Foundation for Statistical   Computing, Vienna, Austria. URL https://www.R-project.org/. We added the information in the text.

The examination of the scatterplots of MRI-R2* evaluations vs time suggested a linear relationship between the two variables. So, the Pearson correlation coefficient was used for evaluating the linear association between two variables. The Spearman rank correlation coefficient was also used with similar results.

We added in the text that the association between R2* and time was studied, for each patient, with Pearson’s correlation coefficient, after the visualization of two-way scatterplots of the data.

Q3 Line 150: time was studied

Q4 Line 153: two-sided

R3- R4. we corrected typos, we thank the reviewer

In agreement with the Editor, there will be English revised language after the manuscript will be accepted.

We sincerely thank the reviewers for appreciating our work.